# Efficacy of Modified Treat-and-Extend Aflibercept Regimen for Macular Edema Due to Branch Retinal Vein Occlusion: 1-Year Prospective Study

**DOI:** 10.3390/jcm9082360

**Published:** 2020-07-23

**Authors:** Yusuke Arai, Hidenori Takahashi, Satoru Inoda, Shinichi Sakamoto, Xue Tan, Yuji Inoue, Satoko Tominaga, Hidetoshi Kawashima, Yasuo Yanagi

**Affiliations:** 1Department of Ophthalmology, Jichi Medical University, 3311-1 Yakushiji, Shimotsuke-shi, Tochigi 329-0431, Japan; r1003ya@jichi.ac.jp (Y.A.); r1208is@jichi.ac.jp (S.I.); r1136ss@jichi.ac.jp (S.S.); yino-tky@umin.ac.jp (Y.I.); notani_s_0906@yahoo.co.jp (S.T.); hidemeak@khaki.plala.or.jp (H.K.); 2Japan Community Health Care Organization Tokyo Shinjuku Medical Center, Tokyo 162-8543, Japan; tanxue1201@hotmail.com; 3Department of Ophthalmology, Graduate School of Medicine, University of Tokyo, Tokyo 113-8655, Japan; 4Department of Ophthalmology, Asahikawa Medical University, Asahikawa 078-8802, Japan; yanagi.yasuo@icloud.com; 5Medical Retina, Singapore National Eye Centre, Singapore 168751, Singapore; 6Medical Retina, Singapore Eye Research Institute, Singapore 168751, Singapore; 7The Ophthalmology & Visual Sciences Academic Clinical Program, Duke-NUS Medical School, National University of Singapore, Singapore 169857, Singapore

**Keywords:** branch retinal vein occlusion, modified treat-and-extend regimen, intravitreal aflibercept

## Abstract

Purpose: To assess the efficacy and safety of a modified treat-and-extend (mTAE) regimen of aflibercept for macular edema (ME) due to branch retinal vein occlusion (BRVO). Methods: This prospective multicentre intervention study evaluated 50 eyes of 50 patients enrolled from October 2016 to September 2017. The patients received intravitreal aflibercept (IVA) injections on an mTAE regimen for a total of 12 months. The main outcome measures were best-corrected visual acuity (BCVA) and central subfield thickness (CST) at 12 months. Results: The baseline BCVA and CST were 0.33 (0.27) and 488 (171) µm (mean (standard deviation)), respectively. The BCVA and CST were significantly improved at month 12 (0.067 (0.19) LogMAR and 295 (110) µm; both *p* < 0.0001, paired *t*-test). The mean number of clinic visits and IVA injections was 6.71 (1.41) and 4.26 (0.71), respectively. The time to first recurrence from the first injection was most frequently 3 months. Conclusion: The mTAE regimen of IVA injections for ME due to BRVO effectively improved BCVA and reduced CST, and thus might be an effective therapy to reduce the number of injections and visits.

## 1. Introduction

Branch retinal vein occlusion (BRVO) is a commonly occurring vascular condition of the retina and may lead to significant loss of vision [1,2,3]. The most frequent causes of vision loss in BRVO are macular edema (ME) and ischaemia [4,5]. Vascular endothelial growth factor (VEGF) is one of the major proteins that is upregulated in BRVO [6], and anti-VEGF therapies have been used as a first-line treatment for ME in eyes with BRVO for the last decade. Several early large-scale studies used an initial six monthly injections of anti-VEGF agents. For example, the first prospective randomised multicentre study, the Branch Retinal Vein Occlusion (BRAVO) trial [7], used monthly intravitreal ranibizumab (IVR) until month 6, and the double-blind, active-controlled, randomised, phase III VIBRANT study [8] used monthly intravitreal aflibercept (IVA) until month 6. The BRIGHTER study [9], which reported that ranibizumab with or without adjunct laser treatment was superior to laser treatment alone in improving mean best-corrected visual acuity (BCVA) from the baseline at 6 months, employed a regimen of at least three initial monthly ranibizumab injections until the stabilisation of visual acuity (VA), followed by VA-based pro re nata (PRN) dosing. However, in real clinical settings, investigators are trying to reduce the number of injections while maximising the visual outcome. The efficacy of IVR injections administered according to a one injection plus PRN regimen has been reported in two studies. Miwa et al. [10] reported that changes in VA and central foveal thickness from the baseline in a group that received one initial IVR injection were not significantly different from those in a group that received three monthly IVR injections. Kawamura et al. [11] reported a mean Early Treatment Diabetic Retinopathy Study (ETDRS) letters score gain of 15.2 letters and central retinal thickness reduction of 230 µm, with 90% of patients having a Snellen equivalent BCVA of 20/40 or better and 15% having a Snellen equivalent BCVA of 20/20. However, such reactive treatment has a potential risk of undertreatment, as has been demonstrated in studies for exudative age-related macular degeneration (AMD).

There have been several attempts to develop an individualised regimen. Among such regimens, the treat-and-extend (TAE) regimen aims to reduce the burden of frequent treatments, specifically the number of visits [12,13]. The TAE regimen was reported to be the most commonly applied regimen of anti-VEGF treatment for AMD [12,13,14,15,16,17]. There have also been several reports on the efficacy of a TAE regimen for ME due to BRVO [18,19]. However, the period until the recurrence of ME varies from individual to individual, and although some patients may need monthly injections, a substantial proportion of them need only one intravitreal injection [20]. Thus, multiple initial intravitreal injections may constitute overtreatment for some patients who would probably need only one injection if a one injection plus PRN regimen was employed. Therefore, we attempted to reduce the number of initial doses for the TAE regimen for ME due to BRVO to avoid overtreatment.

In this study, the second injection was administered based on monitoring every 4 weeks for recurrence of ME. Thereafter, the injection interval was adjusted accordingly. This modified TAE (mTAE) regimen was examined prospectively to evaluate its efficacy and safety in patients with BRVO. The primary efficacy outcome measures were BCVA and central subfield thickness (CST) at month 12, after the initial injection.

## 2. Materials and Methods

### 2.1. Patients and Approval

This prospective multicentre intervention study involved consecutive patients with treatment-naïve ME due to BRVO. The study was initiated in October 2016, and the last patient completed the 1-year study in September 2018. The study protocol was approved by the institutional review board of Jichi Medical University (B18-003). The study followed the tenets laid out in the Declaration of Helsinki. Informed consent was obtained from all patients. The study was registered in the University Hospital Medical Information Network Clinical Trials registry prior to study commencement (27/10/2016, UMIN000024587).

### 2.2. Methods and Ophthalmological Examination

This study was performed at 6 sites in Japan (Institution A, Jichi Medical University Hospital; B, Japan Community Health Care Organization Tokyo Shinjuku Medical Center; C, Ohkubo Eye Clinic; D, Takahashi Eye Clinic; E, Saito Eye Clinic; F, Aoki Eye Clinic). The inclusion criteria were patients aged 20 to 89 years, diagnosed with ME that was central edema secondary to BRVO, within 12 months before screening. The exclusion criteria were a history of previous anti-VEGF treatment, intravitreal or sub-Tenon corticosteroid therapy and/or vitrectomy, fundus photocoagulation treatment or intraocular surgery (without YAG capsulotomy) within 3 months prior to study enrolment, other disease that could cause decreased visual acuity (except moderate cataract), and a history of thromboembolic events within 3 months prior to study enrolment. Although eyes with macular BRVO were included, eyes with hemi-central retinal vein occlusion were excluded.

All patients underwent a complete ophthalmic examination including BCVA assessment with refraction using the 5 M Landolt C VA chart, intraocular pressure measurement, and CST evaluation using spectral-domain optical coherence tomography (OCT), swept-source OCT, indirect ophthalmoscopy, and fundus photography at every visit. The OCT devices and fundus cameras used are listed in Appendix A. Fluorescein angiography (FA) was performed before the initial injection in cases for whom the investigator deemed FA was necessary. The diagnosis of ME was based on OCT and fundus findings. Additional FA was performed at month 12 where the investigator deemed necessary.

### 2.3. mTAE Regimen

Patients were treated with aflibercept according to the mTAE regimen. Initial dosing was not employed. After the first injection, all patients had follow-up visits every 4 weeks. If there was any exudative lesion in the macula (ME and/or serous retinal detachment (SRD) in any of the serial macular OCT scans), the second injection was given, and then the TAE process was begun. At that time, there were no CST criteria. The second injection criterion was any exudative lesion in any of the serial macular OCT scans. After the second injection, if there was no exudative lesion in any of the OCT scans, the eye was defined as having dry macula, IVA was given, and the period to the next treatment was extended by 4 weeks at a time (no maximum interval specified). If there was increased exudative change in OCT findings at the first recurrence, an injection was given, and the visit interval was shortened by 4 weeks; if the exudative change was the same or less compared with the first recurrence, 1 injection was given, and the interval to the next visit remained unchanged. Patients who maintained dry macula after the first injection were followed up every 4 weeks until week 16, and the visit interval could be extended to 3 months after week 16. When the first recurrence of exudative changes was observed after week 16, the TAE regimen was recommended at 3-month intervals; for patients who did not have a recurrence, they were observed every 3 months only. Below we show an example of a specific mTAE regimen (Figure 1a–c). Moreover, we show examples of the administration method for the PRN and TAE regimen in comparison with the mTAE regimen. (Figure 2a,b).

### 2.4. Outcome Measures

The main outcome measures were the mean change in BCVA and CST at month 12. We also examined the numbers of IVA and clinic visits during the 1-year period. The BCVA and CST changes in macular BRVO and major BRVO were also assessed. Macular BRVO was defined as occlusion of only the vein inside the arcade vessels. Major BRVO was defined as all BRVO with ME, other than macular BRVO. Major BRVO is caused by the occlusion of one of the four major branch retinal veins. It involves the entire segment of the retina drained by the vein extending all the way to the peripheral retina. Macular BRVO is caused by occlusion of one of the veins from only the macular region (the part of the retina between the superior and inferior vascular arcades) [21].

### 2.5. Statistical Analysis

Statistical analysis was performed using JMP Pro software version 14.1.0 (SAS Institute, Cary, NC, USA). Decimal BCVA was converted to logMAR. The BCVA and CST were compared with those at the first injection by using a two-tailed paired t-test. Between months 4 and 11, the last observation carried forward was used to impute the missing data.

## 3. Results

### 3.1. Baseline Characteristics

A total of 50 eyes in 50 patients (24 men, 26 women) were included in this study (Table 1). The mean age (standard deviation) of all patients at the start of treatment was 66 (12) years (range 42–85 years). The mean baseline BCVA (logMAR) was 0.33 (0.27), and the mean baseline CST was 488 (171) µm. Of these 50 patients, four dropped out of the study (three were lost to follow-up, and one withdrew consent). Major BRVO was seen in 29 eyes, and macular BRVO was seen in 17 eyes.

### 3.2. VA Outcomes and CST

The mean BCVA improved significantly at months 1, 6, and 12 (0.11 (0.15), 0.033 (0.14), and 0.067 (0.19), respectively) compared with the baseline (0.33 (0.27); all *p* < 0.0001, paired t-test) (Figure 3a,b). The CST decreased significantly after IVA injections at months 1, 6, and 12 (246 (36), 267 (55), and 295 (110) µm, respectively) compared with the baseline (488 (171) µm; all *p* < 0.0001, paired t-test) (Figure 3c).

### 3.3. Mean Numbers of IVA Injections and Clinic Visits

The mean number of IVA injections was 4.26 (0.71). Seven eyes received only one injection (Figure 4a). The mean number of clinic visits was 6.71 (1.41) in the 12-month observation period (Figure 4b).

### 3.4. Time to First Recurrence from First Injection

The time to first recurrence from the first injection is shown in Figure 4c. Six eyes showed recurrence of exudative changes at month 2. Recurrence occurred most frequently at month 3 (15 eyes, 33%). First recurrence occurred at month 7 in three eyes, two of which had no significant change in BCVA at month 7 compared with the previous visit. In one out of the three eyes, BCVA decreased from −0.1 to 0.1 (logMAR). Seven eyes showed no recurrence at all, of which five eyes had major BRVO, and two eyes had macular BRVO.

Recurrence was seen in six eyes at month 1. Five out of these six eyes showed repeated recurrence, despite injection every 4 weeks; however, one out of these eyes showed no recurrence during the TAE period.

### 3.5. Comparison between Major BRVO and Macular BRVO

The comparison between major BRVO and macular BRVO is shown in Table 2. Figure 5 shows the improvement of BCVA and CST in patients with major BRVO (29 eyes) and macular BRVO (17 eyes). The mean BCVA in major and macular BRVO significantly improved at month 12 (0.071 (0.19) and 0.067 (0.19), respectively) compared with the baseline (0.33 (0.25) and 0.34 (0.30), respectively; both *p* < 0.0001, paired *t*-test).

The mean BCVA in major BRVO and macular BRVO had improved significantly at month 12 (0.071, 0.067, respectively) compared with the baseline (0.33, 0.34, respectively, *p* < 0.0001, paired t-test). The mean CST in major BRVO and macular BRVO had improved significantly at month 12 (304, 282 µm, respectively) compared with the baseline (509, 477 µm, respectively, *p* < 0.0001, paired *t*-test). 

The mean CST in major and macular BRVO significantly decreased at month 12 (304 (129) and 282 (70) µm, respectively) compared with the baseline (509 (161) and 477 (191) µm, respectively; both *p* < 0.0001, paired *t*-test). The mean numbers of IVA injections showed no significant difference between major and macular BRVO (4.24 and 4.29, respectively; *p* = 0.93, paired *t*-test). The mean number of visits was significantly less in macular BRVO than in major BRVO (6.97 and 6.29, respectively; *p* = 0.01, paired *t*-test).

### 3.6. Safety Outcomes through Month 12

No serious ocular complications associated with IVA injections were observed. No serious systemic adverse events were noted.

## 4. Discussion

In this study, we demonstrated the effectiveness of the mTAE regimen of IVA injections for treating ME due to BRVO in improving visual acuity and reducing CST. The mean baseline BCVA (logMAR) was 0.33 and improved to 0.067 at month 12. The mean baseline CST was 488 µm and improved to 295 µm at month 12. The mean number of injections and clinic visits was 4.26 (0.71) and 6.71 (1.41), respectively. The BCVA and CST values were consistent with those in previous reports [7,8,10,11,18,19], and the mean number of injections was similar to that of the one injection plus PRN regimen [10,11]. The mean number of clinic visits was similar to that in the TAE regimen [18,19]. Many previous studies investigated PRN and TAE regimens for BRVO [7,8,10,11,18,19,22], but this mTAE regimen is the first to combine the best features of PRN and TAE regimens.

To our knowledge, there has been no report of a TAE regimen using aflibercept for ME due to BRVO. Previous studies described TAE regimens using ranibizumab [18,19]. Although the change in mean visual acuity was similar to that in our study, the mean number of injections was higher. Although direct comparison is difficult, it is plausible that the difference in the number of injections could be attributed to the different regimens employed between their study and ours.

A study by Miwa et al. [10] used the one injection plus PRN regimen, and, although the mean number of visits was higher in their study than in ours, the change in mean visual acuity (logMAR) from the baseline at month 12 was similar to our finding. Furthermore, Campochiro et al. [23] reported that 50% of patients required intravitreal ranibizumab injections 4 years after the first treatment. Another study used a “treat and monitor” or “inject and observe” regimen, i.e., a modified PRN regimen without a loading dose [24]. However, we think that the PRN regimen places a greater burden on both healthcare workers and patients. The present study clearly shows that the mTAE regimen can reduce the mean number of visits compared with the PRN regimen.

In this study, the time of the first recurrence varied, highlighting the need to individualise the administration method. Hikichi et al. [20] also reported a TAE regimen that had an initial loading dose of three consecutive monthly injections, but it might result in overtreatment in most cases. However, some patients might need an initial loading dose of several consecutive monthly injections, as shown in the present study. In the current protocol, if any exudative changes (cystoid ME or SRD) were noted at month 1, the TAE regimen was started from month 2, and the treatment interval was adjusted thereafter [25]. Further consideration of the need for the initial dosing (until the exudative lesions disappear) is warranted, because patients with ME or SRD at month 1 may be overtreated.

There was no significant difference in the mean number of injections and visits between major and macular BRVO in the present study, during the 12-month observation period. Hayreh et al. [21] reported that major and macular BRVO were two markedly distinct conditions. They showed that the resolution time for retinopathy of macular BRVO was 4 years in terms of the natural history, while that for major BRVO was 1.5 years. Moreover, a study of a PRN regimen for anti-VEGF drugs showed that the number of anti-VEGF injections in macular BRVO was lower than that for major BRVO [26,27]. Our currently ongoing study will clarify whether the number of injections for macular BRVO might decrease the dosing frequency as compared with major BRVO.

The present study demonstrated changes in VA from the baseline that were similar to those of previously reported PRN or TAE regimens, despite the lower number of visits and the lower number of injections with complications. BRVO is a chronic disease that requires long-term intravitreal injections of anti-VEGF, and severe side effects have been reported. Circulating VEGF protects the integrity and patency of vessels, and prolonged anti-VEGF treatment can lead to an increased risk of thromboembolic events [28,29,30]. Moreover, individuals with BRVO have a significantly increased risk of developing acute myocardial infarction than those without BRVO [31]. It is likely that complications associated with IVA can be reduced by a regimen with a minimal number of injections.

This study has some limitations that should be noted. First, the study was designed as a single-arm trial. Second, the sample size was relatively small. Third, only Japanese patients were included. Fourth, the observation period was relatively short. Therefore, further randomised controlled studies are needed to show that our regimen is non-inferior to the gold standard treatment regimen of the 6-monthly plus PRN regimen.

## 5. Conclusions

In this prospective study, an mTAE regimen of IVA injections for treating ME due to BRVO effectively improved VA and reduced CST. This can be effective not only for improving VA and structural outcomes but also for reducing the number of required injections and hospital visits, and thus might be an effective therapy for decreasing the burden on both healthcare workers and patients.

## Figures and Tables

**Figure 1 jcm-09-02360-f001:**
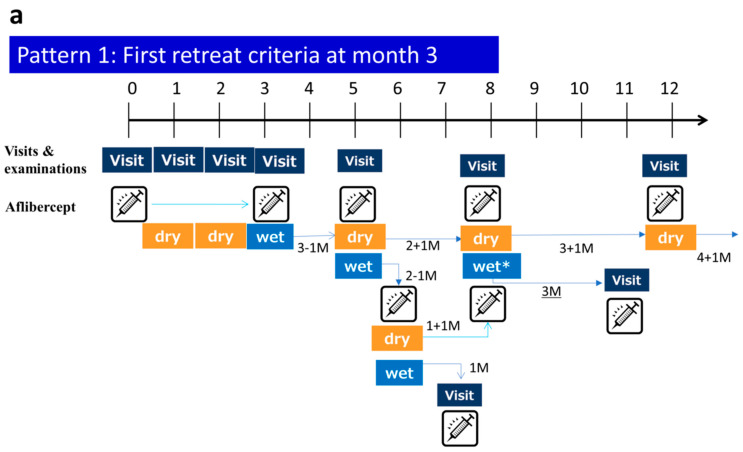
Example of the administration method for the modified treat-and-extend (mTAE) regimen. (**a**) This pattern shows the first retreat criteria met at month 3. (**b**) This pattern shows the first treat-and-extend (TAE) start criteria met at month 1. (**c**) This pattern shows the recurrence that occurred after the end of the monthly examinations.

**Figure 2 jcm-09-02360-f002:**
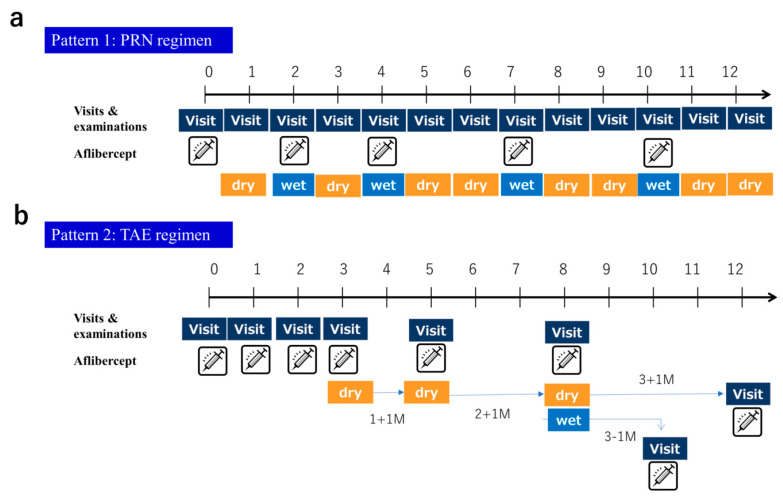
Example of the administration method of the pro re nata (PRN) and TAE regimen compared with the modified TAE regimen. (**a**) This pattern shows the PRN regimen. This pattern increased the number of clinic visits. (**b**) This pattern shows the TAE regimen. This TAE regimen required a loading phase of three consecutive monthly injections.

**Figure 3 jcm-09-02360-f003:**
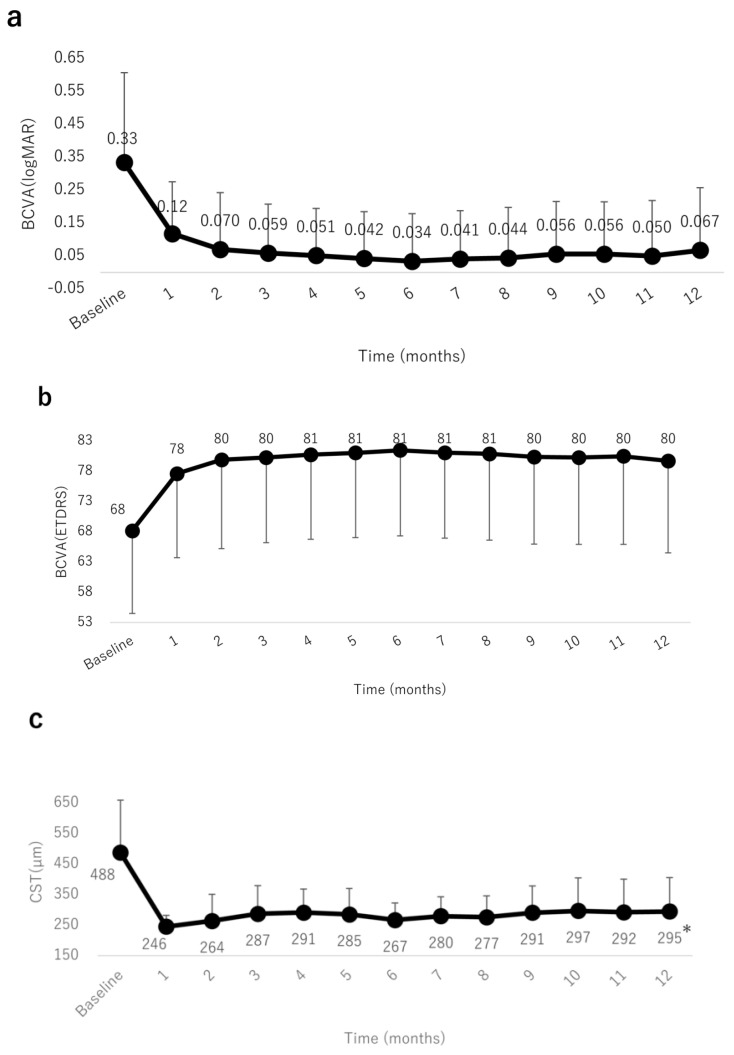
Mean BCVA (logMAR) and mean change from baseline BCVA (Early Treatment Diabetic Retinopathy Study (ETDRS) letter score) over time to month 12. (**a**) The mean BCVA improved significantly at month 1 after intravitreal aflibercept (IVA) injection, and the improvement continued through month 12. (**b**) The mean change from the baseline BCVA significantly increased over time through month 12. (**c**) The mean central subfield thickness (CST) over time to month 12. The mean CST improved significantly at month 1 after IVA injection, and the improvement was sustained through to month 12. *: *p* < 0.0001.

**Figure 4 jcm-09-02360-f004:**
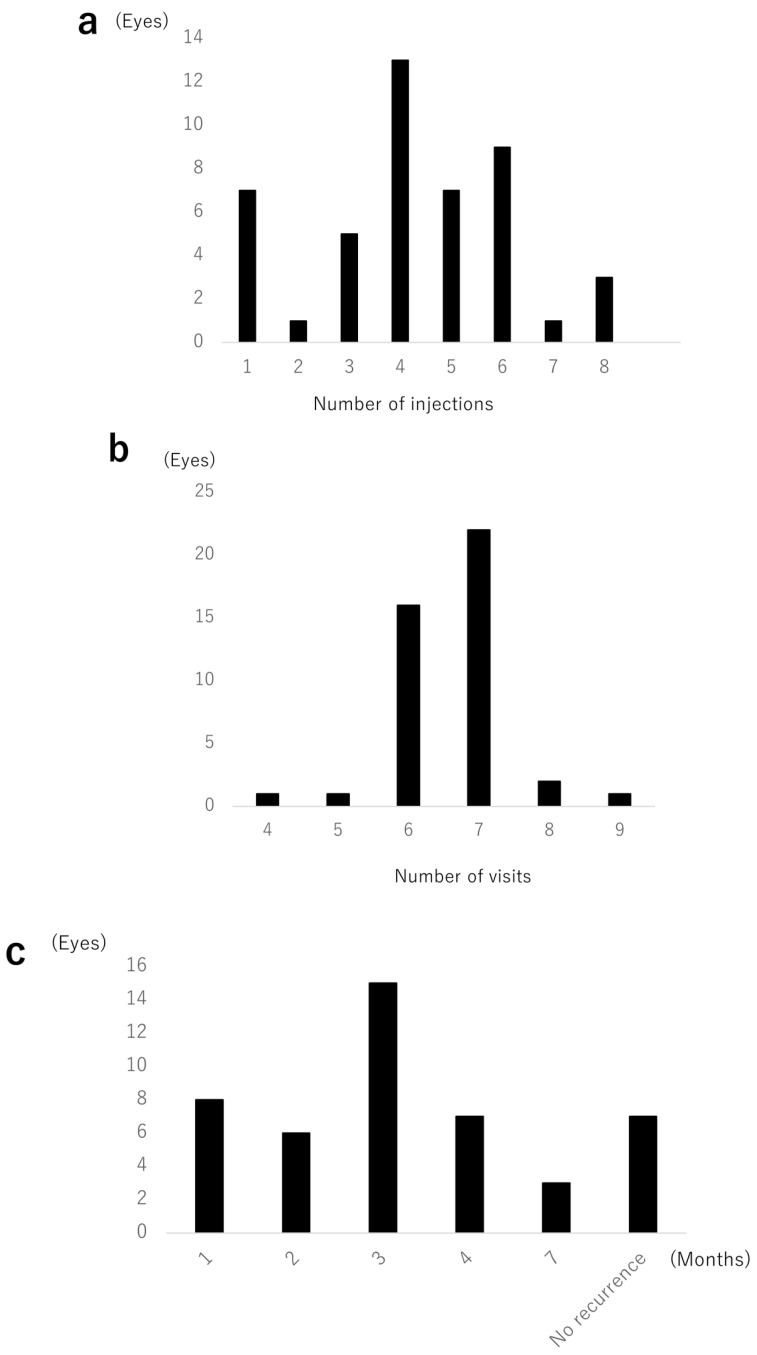
(**a**) Distribution of number of injections. Seven eyes (16%) received only 1 injection. The most frequent number of injections was 4 times (13 eyes, 28%). (**b**) Distribution of number of visits. The minimum was 4, and the maximum was 9 times. The most frequent number of visits was 7 times (22 eyes, 47%). (**c**) The time to first recurrence from the first injection. The first recurrence most frequently occurred at month 3 (15 eyes, 33%). Seven eyes (15%) had no recurrence.

**Figure 5 jcm-09-02360-f005:**
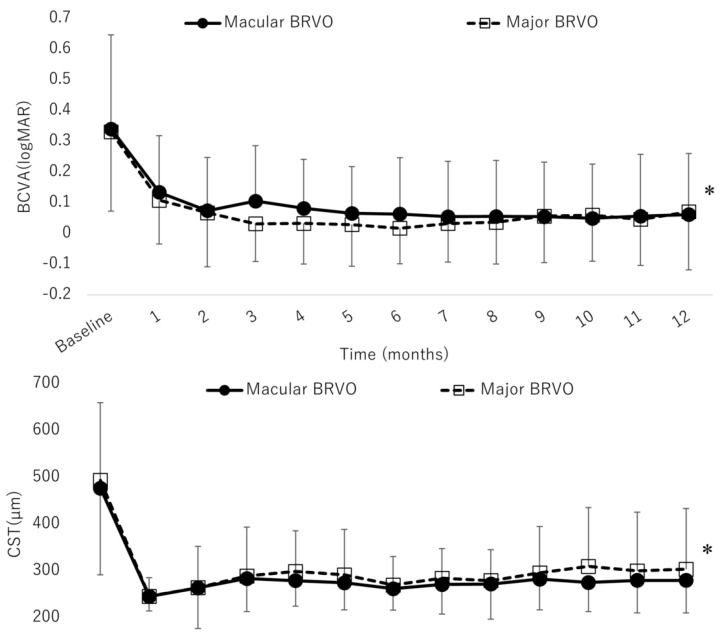
Mean BCVA (logMAR) and CST of major branch retinal vein occlusion (BRVO) and macular BRVO over time to month 12. *: *p* < 0.0001.

**Table 1 jcm-09-02360-t001:** Characteristics of enrolled patients.

Cases	50
Age (years; mean (SD))	66 (12)
Sex (male/female)	24/26
BCVA (logMAR; mean (SD))	0.33 (0.27)
CMT (µm; mean (SD))	488 (171)

BCVA, best-corrected visual acuity; CMT, central macular thickness; SD, standard error.

**Table 2 jcm-09-02360-t002:** Comparison of major BRVO and macular BRVO.

	Major BRVO (n = 29)	Macular BRVO (n = 17)	*p*. Value *
Baseline BCVA (logMAR)	0.33 (0.26)	0.34 (0.31)	0.91
Baseline CRT (µm)	509 (161)	477 (191)	0.55
BCVA at month 12 (logMAR)	0.071 (0.19)	0.067 (0.20)	0.95
CRT at month 12 (µm)	304 (129)	282 (71)	0.54
Mean number of injections	4.24	4.29	0.93
Mean number of visits	6.97	6.29	0.01
Duration between symptoms and initial therapy (month)	1.62	4.95	0.01

Mean (standard deviation), * paired *t*-test.

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
