# Peer review of "Efficacy of Modified Treat-and-Extend Aflibercept Regimen for Macular Edema Due to Branch Retinal Vein Occlusion: 1-Year Prospective Study"

_jcm, 2020, doi:10.3390/jcm9082360_

Round 1

Reviewer 1 Report

This manuscript describes a new protocol for treating ME due to BRVO using aflibercept. The modified treat-and-extend regimen combines the advantages of both PRN regimen and TAE regimen. The manuscript reads well and the research was carried out comprehensively. The topic is worthwhile investigating and will be interesting to the field.

  1. The authors used a modified treat-and-extend regimen and provided some examples in the supplements. However, It will be very helpful for the readers to understand the proposed protocol if there is any schematic figure showing in the main manuscript. The schematic figure should compare the proposed protocol with PRN and TAE.
  2. The order of graphs in Figure 2 is different from the caption. Please correct the order.

Author Response

Response to Reviewer 1 Comments

Point1. The authors used a modified treat-and-extend regimen and provided some examples in the supplements. However, It will be very helpful for the readers to understand the proposed protocol if there is any schematic figure showing in the main manuscript. The schematic figure should compare the proposed protocol with PRN and TAE.

Thank you for your comment. We moved Supplemental Figure S1-S3 to main manuscript. (Figure1a-1c) And, we have added figures which showing protocol with PRN and TAE regimen. (Figure2a,2b) We corrected figure legends and Supplementary information.

.

Point2. The order of graphs in Figure 2 is different from the caption. Please correct the order.

Thank you for pointing this out. We have corrected the order of the graphs in Figure 2.

We have also revised the English in the manuscript and added some references.

Reviewer 2 Report

In this paper, the authors aimed to demonstrate the effectiveness of a modified treat and extend regimen (a fusion between PRN and TAE) with aflibercept for macular edema secondary to BRVO. The results of this prospective longitudinal study (12 months) show improvement in visual acuity and a reduction of CST.

The paper is clear but would benefit from an improvement of awkward phrasing and grammar.

Methods :

Was an initial loading dose performed?

The authors state: ‘After the second injection if there was no exudative lesion in the macula, defined as dry macula’. Are the authors referring to the foveal area or to the whole macular area? The term is very vague. Please clearly define with ETDRS grid.

Discussion:

-please be more concise

The authors modified TAE seems similar to the previously described ‘treat and monitor’ protocol (PMID: 29274022) or ‘inject and observe’. Moreover, these results are consistent with previous results in the literature.

While it is true that ME secondary to BRVO treated by TAE has not been the specific topic of any study, studies such as Picchi et al (PMID: 29274022) and O’Day et al (PMID: 32093666) are very similar.

Author Response

Response to Reviewer 2 Comments

We wish to express our appreciation to the reviewer for their insightful comments, which have helped us significantly improve the paper.

Methods :

Was an initial loading dose performed?

The current modified TAE regimen did not require a loading phase of three consecutive monthly injections. (Page 5, Line115-116)

The authors state: ‘After the second injection if there was no exudative lesion in the macula, defined as dry macula’. Are the authors referring to the foveal area or to the whole macular area? The term is very vague. Please clearly define with ETDRS grid.

If there was no exudative change in all SD-OCT scans after the second injection, we defined it as dry macula. That area includes, but not limited within, ETDRS grid area. We have revised this accordingly (Page 5, Line120).

Discussion:

-please be more concise

 We have revised the Discussion section accordingly.

The authors modified TAE seems similar to the previously described ‘treat and monitor’ protocol (PMID: 29274022) or ‘inject and observe’. Moreover, these results are consistent with previous results in the literature.

While it is true that ME secondary to BRVO treated by TAE has not been the specific topic of any study, studies such as Picchi et al (PMID: 29274022) and O’Day et al (PMID: 32093666) are very similar.

We think that the treat-and-monitor protocol (PMID: 29274022) is similar to the one plus PRN regimen, and does not contribute to reduce the number of clinic visits. This modified TAE regimen aims to reduce the number of both intravitreal aflibercept injections and clinic visits. We have modified discussions as such (Page 9, Line 223 to 227).

We have also revised the English in the manuscript and added some references.
